# New Frontiers in Peripheral Nerve Regeneration: Concerns and Remedies

**DOI:** 10.3390/ijms222413380

**Published:** 2021-12-13

**Authors:** Polina Klimovich, Kseniya Rubina, Veronika Sysoeva, Ekaterina Semina

**Affiliations:** 1National Cardiology Research Center Ministry of Health of the Russian Federation, Institute of Experimental Cardiology, 121552 Moscow, Russia; lex2050@mail.ru (P.K.); e-semina@yandex.ru (E.S.); 2Faculty of Medicine, Lomonosov Moscow State University, 119991 Moscow, Russia; veroniks@mail.ru

**Keywords:** peripheral nerve regeneration, axon growth, Schwann cells, exosomes, microRNA, multipotent stromal cells

## Abstract

Topical advances in studying molecular and cellular mechanisms responsible for regeneration in the peripheral nervous system have highlighted the ability of the nervous system to repair itself. Still, serious injuries represent a challenge for the morphological and functional regeneration of peripheral nerves, calling for new treatment strategies that maximize nerve regeneration and recovery. This review presents the canonical view of the basic mechanisms of nerve regeneration and novel data on the role of exosomes and their transferred microRNAs in intracellular communication, regulation of axonal growth, Schwann cell migration and proliferation, and stromal cell functioning. An integrated comprehensive understanding of the current mechanistic underpinnings will open the venue for developing new clinical strategies to ensure full regeneration in the peripheral nervous system.

## 1. Introduction

The peripheral nervous system (PNS) exhibits a limited capacity for functional and morphological repair and regeneration. Peripheral nerve recovery is a multistep process with a complex molecular and cellular regulatory circuitry. Severe injury of peripheral nerves often results in a loss of motor, sensory, and autonomic functions of innervated organs and tissues, therefore calling for novel treatment strategies to ensure effective regeneration.

This review aims to augment the classical view on the basic mechanisms of nerve regeneration with the recent experimental data on exosome-transferred microRNAs contributing to axonal growth and Schwann cell functioning. First, we overview the role of neurotrophins, guidance receptors, extracellular matrix, and adhesion proteins as well as immune system contribution to nerve regeneration. The recently emerging data expand the well-known players in nerve regeneration, thus opening new avenues in tackling this problem efficiently.

## 2. Molecular Basis for Nerve Regeneration 

Since mechanisms orchestrating axon regeneration are complicated and diverse, it is important to take into consideration all of them before developing new strategies to improve regeneration and reinnervation. At the site of injury, a subset of cellular and biochemical responses in neurons occur, both in the cell somas and axons, and in their innervated organs. The damage to the axonal membrane after nerve lesion leads to the rapid sodium and calcium influx into the axoplasm and generates a high-frequency burst of action potentials at the proximal part of the axons [1]. As a result, several protein kinases, including calcium/calmodulin-dependent kinase 2 (CMAK2), protein kinase A (PKA), protein kinase C (PKC), and mitogen-activated protein kinase (MAPK) such as Erk1 and Erk2, c-jun N-terminal kinase (JNK) and p38 kinases are activated [1,2]. Active kinases are transported into the cell somas via retrograde trafficking system, where they modulate signaling pathways resulting in further activation of several transcription factors such as cAMP responsive element binding protein (CREB), activating transcription factor 3 (ATF-3) and c-Jun [3,4]. The combination of these processes enables the onset of gene transcription programs associated with nerve regeneration and known as RAGs (regeneration associated genes) [5]. 

RAG programs ensure the change of neuron phenotype from neuro-transmitter state conducting the action potential to the pro-regenerative one [6]. In particular, in regenerating neurons, the elevated expression of genes encoding cytoskeleton proteins (actin and tubulin), GAP43 protein (growth-associated protein-43), neurotrophic growth factors (NGF and BDNF), and their receptors Trk [7], transcription factors (Sox11, c-Jun) [8], cell adhesion molecules, and guidance receptors has been detected, while gene expression of neurofilaments [9], neuropeptides, and neurotransmitters have been inhibited [10]. This gene expression program sustains the formation of new growth cones and axon elongation required for nerve regeneration.

Along with that, the distal stump of the injured nerve undergoes a set of molecular and cellular changes described as Wallerian degeneration. Immediately after injury, both axons and myelin at the distal stump degenerate. The surviving Schwann cells as well as macrophages migrating into the injury site clear the debris. Schwann cells engulf the degrading myelin contributing to debris clearance and repurpose it for future remyelination [6]. Loss of contact with axons and critical signals such as neuregulin-1 triggers Schwann cell transdifferentiation resulting in a change of their myelinating phenotype towards repair-cell phenotype (regenerating and proliferating cells) [11]. Having lost contact with their axons, Schwann cells decrease the content of structural proteins, such as myelin basic protein and myelin-associated glycoprotein. In contrast, the expression of nerve growth factor (NGF), basic fibroblast growth factor (bFGF), neurotrophin 3 (NT-3), receptor p75, glia maturation factor-β (GMF-β), neural cell adhesion molecule (NCAM), and L1 and other molecules, which stimulates regeneration, becomes up-regulated in Schwann cells ensuring axonal growth through the tissue damaged area [12].

The joint action of Schwann cells and macrophages results in debris removal after Wallerian degeneration. Schwann cells actively proliferate at the distal stump and align in columns called bands of Büngner. These columns create a permissive environment rich in trophic factors and ensuring guided axonal growth [13].

Inflammation is another well-recognized aspect of nerve regeneration, since the extent and duration of inflammation affects the regeneration ultimate outcome. The balance between M1 and M2 polarization is extremely important for peripheral nerve regeneration, since the shift towards M1 state and the lack of M2 macrophages lead to a chronic inflammation and impaired regeneration [14]. M2 macrophages are central mediators of nerve regeneration rooted in their ability to respond to injury-induced hypoxia. Hypoxia triggers HIF-1α (hypoxia-inducible factor 1 subunit alpha) production in macrophages, followed by an increased expression of VEGF-A (vascular endothelial growth factor A) being a key regulator of endothelial cell proliferation and migration [15]. In turn, macrophage-induced angiogenesis guides the trajectory of Schwann cell migration and the formation of Büngner bands. Additionally, macrophages promote cell proliferation and migration, as well as contributing to growth-factor production and removal of apoptotic cells [16,17]. 

Elongation of the injured axon in the proximal stump of the regenerating nerve occurs due to the transformation of the axon tip into a highly mobile structure with a so-called “growth cone” at the end. Growth cones guide the axons towards their target organs along the gradient of signals produced by the microenvironment, such as neurotrophic growth factors, extracellular matrix (ECM) proteins, guidance receptors, and cell adhesion molecules. Schwann cells, macrophages, and target organs act as a source of neurotrophic and growth factors [6]. In particular, CNTF (ciliary neurotrophic factor), secreted by Schwann cells, stimulates growth and regeneration of the damaged axons. CNTF is a key factor that promotes phosphorylation of transcription factor STAT3, which is retrogradely transported into neuron somas [18]. Activation and nuclear translocation of STAT3 is required for axon elongation and neuron protection against apoptosis [19]. bFGF and NT-3, derived from target organs, are also retrogradely transported along the axons into neuronal somas. Overall, on entering the neuronal somas these factors promote secondary messengers’ production (cAMP and cGMP), which, in turn, activates MAPK signaling cascades and stimulates axonal growth [18].

Overall, the molecular mechanisms implicated in axonal regeneration and pathfinding after injury are multifactorial. Successful regeneration is dependent upon a crosstalk between the growing axons, Schwann cells and other cells, timely revascularization, and stimuli produced by the target organs. This interaction is coordinated by signals derived from transmembrane receptors, guidance molecules, ECM components, and cell adhesion molecules, which are expressed by all neural cells and act in a paracrine mode. In addition to these canonical mechanisms of intercellular interaction, the possibility of communication through secreted exosomes has been recently explored. The pertinent literature points to the exosome-mediated lateral transfer of regulatory molecules between Schwann cells and neurons important for axonal growth and regeneration [20]. The role of exosomes in nerve regeneration will be discussed in detail below.

### 2.1. Cell Adhesion Molecules and Extracellular Matrix Proteins in Peripheral Nerve Regeneration

Peripheral nerve repairment relies on the ability of regenerating axons to navigate through a structurally altered ECM, myelin, and cell debris, infiltrating inflammatory cells. During early Wallerian degeneration, not only do the axons degenerate and Schwann cells adopt transdifferentiation, but ECM undergoes a substantial degradation thus removing the inhibitory influence of chondroitin-sulfate proteoglycans and myelin-associated glycoprotein, predominantly expressed in the matrix of intact axons, and facilitating the new axon sprouting [21]. The spatio-temporally relevant degradation of ECM provided by the activated proteases is the limiting factor of regenerating axonal growth. Pro-inflammatory cytokines such as HIF-1 and TNF-α induce secretion of plasminogen activators (PAs) and matrix metalloproteinases (MMPs) by neurons, Schwann cells, and macrophages involved in matrix remodeling [22,23].

Following ECM degradation, Schwann cells and fibroblasts rapidly up-regulate the production of new ECM and basement membrane proteins at the site of injury, with laminin, fibronectin, and collagen being predominant types promoting nerve regeneration. This change in the ECM composition promotes the axon growth-cone formation and axon elongation with new integrin-mediated adhesive contacts playing an important role in regenerating peripheral axons. In particular, cytoskeleton proteins undergo qualitative and quantitative changes characterized by an increased expression of actin, tubulin, and peripherin, and a decrease in neurofilaments regulating the axon caliber [24]. The mobile path-finder parts of the growth-cone filopodia and lamellipodia are constantly searching for a permissive substrate and, once encountering the growth-cone receptors, integrins bind to ECM and form adhesions. Integrins are heterodimeric receptors consisting of α and β subunits with a unique affinity of each integrin receptor for different ECM proteins. Specifically, type I, III, and V collagens interact with α1β1 and α2β1 integrins, while α4β1, α5β1, and α8β1 integrins mainly bind fibronectin [25]. 

Using an in vivo model, we have demonstrated that the increase in α5β1 integrin expression is a key feature of peripheral nerve regeneration after injury. Our in vitro experiments revealed that the increase in α5β1 integrin expression stimulates neuritogenesis, while α5β1 integrin inhibition disrupts neurite outgrowth on fibronectin matrix [26]. The interactions between ECM substrates and integrin receptors alter the intracellular level of second messengers (cAMP) in the growth cone resulting in activation of downstream effector proteins such as focal adhesion kinase (FAK) and integrin-linked kinase (ILK), which further activate small GTPases, leading to cytoskeleton remodeling and an increased mobility of the growth cone [25]. Accordingly, integrin receptors are coupled to F-actin fibers; F-actin polymerization pushes the plasma membrane forward at the protrusion domain of the growth cone, while the stabilized microtubules polymerize, relying on the actin network [27,28]. 

In addition to cell adhesion molecules, the elongation of the growth cone and the trajectory of axon growth regeneration are controlled by guidance receptors being expressed on both neurons and Schwann cells.

### 2.2. Guidance Molecules in Nerve Regeneration

Recent decades have witnessed substantial progress in describing the guidance molecules and related mechanisms required to transmit their signals in embryogenesis. Studies on peripheral nerve regeneration in adults have revealed a key role of signaling cascades downstream of the guidance receptors, such as EphrinB2, Slit3, and Netrin1 [29]. After the peripheral nerve injury, Netrin-1 expression is significantly elevated in Schwann cells migrating from the distal stump, in the cell somas of sensory neurons, and in the axons of both motor and sensory neurons. Netrin-1 interacts with Neogenin on Schwann cells and CD146 receptor on endothelial cells, boosting their proliferation and migration. Netrin-1 can also contribute to axon elongation through interaction with receptors DCC (Deleted in Colorectal Cancer) along with other molecules, such as Neogenin, DSCAM and CD146 on the growth cone of regenerating axons. Interestingly, Netrin-1 can act as a bifunctional axon guidance cue: Netrin-1-DCC binding results in axon attraction, while Netrin-1-Unc5A–D (Uncoordinated receptor 5A–D) interaction leads to axon repulsion and slows down the rate of axon growth during regeneration, adding further complexity to the regulatory circuitry [30]. 

Schwann cells migrating from the injured nerve stumps and expressing EphB2 come into a direct contact with fibroblasts expressing EphrinB2. This interaction results in Ephrin-B/EphB2-mediated cell sorting of these two types, which in turn coordinates migration of Schwann cells as multicellular Büngner bands and guides the axonal growth across the injury site [31].

Macrophages secrete high levels of Slit3, while the migrating Schwann cells and fibroblasts inside the nerve bridge as well as the regenerating axons express Robo1 receptor on their surface. The underlying mechanism of Slit3 action, being a restrictive signal to stay inside the nerve bridge for Robo1-expressing growing axons and migrating Schwann cells, reveals an important function of Slit3-Robo1 signaling axis for axon regeneration. Aligning with this pattern of Slit3 and Robo1 expression, numerous axon regeneration and cell migration defects have been observed in recovering nerves of Sox2-, Slit3-, and Robo1-deficient mice [32].

Semaphorins are well-described chemorepulsive proteins in the developing nervous system regulating axon pathfinding, yet they fall short of studies as far as their role in PNS recovery after injury is concerned. Sema3A is produced by fibroblasts in the epineurial space and appears to be secreted into the extracellular matrix. Sema3A binds NRP-1 (neuropilin-1 receptor) on the membrane of axon growth cones, which leads to their collapse, the change of the trajectory of outgrowing axons and impaired reinnervation of the target organs [33]. 

Along with the canonical guidance molecules, literature provides abundant data on glycosylphosphatidylinositol (GPI)-anchored proteins also playing a regulatory role in axon pathfinding. Peering into the functioning of urokinase receptor (uPAR), a (GPI)-anchored protein, in nerve regeneration, we obtained solid evidence for a unique role of uPAR in stimulating axonal growth and proliferation of Schwann cells in response to injury in PNS [34]. Using an in vitro model of a Neuro2a cell line differentiated into neurons, we have demonstrated that uPAR expression in axons is crucial for neurite outgrowth. Moreover, exploiting the DRG (dorsal root ganglia) ex vivo explant model, we have shown that uPAR plays an important role in axon branching and neural cell migration. It is well-known that active urokinase can cleave uPAR, which results in uPAR shedding into culture supernatants and human body fluids. Soluble uPAR (suPAR) can function as a ligand for FPRL1 (formyl peptide receptor 1) a chemokine receptor in neurons, “attract” and guide the growing axons along suPAR gradient [35].

### 2.3. Neurotrophins and Cytokines in Peripheral Nerve Regeneration

Neurotrophic factors are a heterogeneous group of signaling molecules that play a pivotal role in PNS development, maintenance, and plasticity. The group comprises three multigene families—the neurotrophins (NGF, BDNF, NT-3, and NT-4/5), neuropoetic cytokines (IL-6, IL-11, leukemia inhibitory factor (LIF), oncostatin M (OSM), CNTF), and the glial cell line-derived neurotrophic factor ligands (GDNF, neurturin, artemin, and persephin) [36].

Previous studies have shown that growth factors and cytokines are intricately involved in the axon regeneration, formation of Büngner bands, and neuronal survival [37]. When sensory and motor neurons are injured, several growth neurotrophic factors and cytokines are up-regulated or activated in neurons and the surrounding tissues. The change in the expression pattern largely relies on the specific type of the damaged neurons. In particular, it has been revealed that NGF and BDNF expression is up-regulated in sensory neurons [38] and NT-3 in motoneurons [39], while GDNF exert a pronounced trophic effect on sensory neurons [40]. Along with that, the expression of TrkA and TrkB, which are high-affinity receptors for neurotrophins NGF, BDNF, and NT-4, correspondingly, is elevated on the growth cone of the regenerating axons [41]. BDNF and GDNF expression has been reported to be also augmented in Schwann cells. Interestingly, while Schwann cells do not express high-affinity receptors, they do express a low-affinity receptor p75—a common receptor for all neurotrophins (NGF, BDNF, NT3, NT4) [37].

Multiple studies indicate that the absence of endogenous BDNF impedes the growth of sciatic nerve axons and their myelination after injury [42,43], whereas local delivery of recombinant BDNF using a prolonged release capsule system significantly improves nerve regeneration [44]. We have recently developed a therapeutic approach using the local delivery of a genetic construct for a combined expression of BDNF and urokinase ensuring a long-term expression of these proteins at the site of nerve injury [45]. The abovementioned strategy is less stressful compared to serial protein injections. Moreover, the combined action of these two factors—BDNF, stimulating the axon growth, and urokinase, being involved in ECM remodeling—ensures a complex effect on morphological and functional nerve recovery [34,46]. 

Several papers reported that the sustained delivery of GDNF, rather than NGF, to the site of nerve injury enabled efficient in vivo regeneration of both sensory and motor axons [47]. Using an ex vivo three-dimensional model of mouse DRG explants, we have shown that GDNF delivery exerts a more pronounced stimulatory effect on axonal growth and regeneration as compared to BDNF or NGF, as well as promotes glial cell migration [37,48].

Non-neurotrophic growth factors also actively participate in peripheral nerve regeneration. VEGF secreted by Schwann cells promotes the ingrowth of newly formed blood vessels into the area of nerve injury [49], while bFGF secreted by fibroblasts stimulates the proliferation of Schwann cells, therefore ultimately facilitating tissue restoration [27]. 

Schwann cells are the main source of neuropoetic cytokines such as LIF, CNTF and IL6 at the site of injury [36]. Specifically, an increase in CNTF expression in Schwann cells using adeno-associated vectors (AAV) for gene therapy promoted nerve regeneration after sciatic nerve injury in mice. Viral transduction with CNTF construct stimulated Schwann cell differentiation in vitro and in vivo and secretion of myelin proteins. A significant increase in GAP43 expression in sensory neurons, a marker of axonal regeneration, and improved compound muscle action potential was also reported [50]. Specifically, the released CNTF binds to its receptor expressed preferentially on motoneurons; it exerts a paracrine effect, thus promoting neuronal survival, axonal growth, and sprouting. Interaction of CNTF with the axon growth cones of the injured axons induces STAT3 phosphorylation, which is further retrogradely transported to the neuron somas triggering the gene transcription program of neuronal survival and axon regrowth [27]. 

LIF has a pattern similar to CNTF activation in sympathetic neurons. LIF and CNTF employ comparable signaling pathways, and their roles in reinnervation is considered overlapping. Following the nerve transection, LIF secretion is up-regulated in Schwann cells. In vivo studies imply LIF in promoting axon regeneration of motor neurons, since LIF exogenous delivery after nerve injury augments muscle mass and muscle contraction force. Specifically, regeneration of motor neuron axons is impaired in LIF knockout mice suggesting an essential role of this cytokine in neuronal survival and recovery after injury [27,51]. In addition, IL-6 and its receptor expression is up-regulated in the same type of neurons after the nerve injury and their joint [27]. 

Despite the well-documented stimulatory effects of exogenously delivered neurotrophic factors and cytokines on nerve regeneration obtained through experimental models in vivo and in vitro, their clinical application still has certain limitations. First, their elevated local concentration can impede nerve regeneration through potentially triggering neuron apoptosis. It has been reported that exogenously administrated BDNF in high doses of (20 μg versus 2 μg) can induce neuronal apoptosis at the site of peripheral nerve injury. The underlying molecular mechanisms reside in the preferential binding of BDNF to p75 receptor, but not to TrkB, inducing apoptosis [37]. In line with this, the exogenous delivery of NGF can attenuate axon regeneration after axonotomy of the sciatic nerve, probably by decreasing the responsiveness of neuronal somas to injury [52]. Another suggestion made in this paper points to the NGF effects on axotomized fibers being related not to axon regeneration, but rather to NGF indirect action on Schwann cells [52]. Secondly, growth factors and cytokines with a peptide structure and a short half-life in the blood circulation require a repetitive administration, resulting in a high risk of an increased trauma at the site of injury and pain. Moreover, the cost efficiency of this therapy is also an issue. 

### 2.4. The Role of Exosomes in Regeneration of Peripheral Nerves 

Recent advances in cell and molecular biology highlighted a novel mechanism that ensures intercellular communication via exosomes known to be secreted by all cell types [53,54]. As of today, a large body of evidence indicates that exosomes play a pivotal role in transferring information via their highly selective cargo. Exosomes comprising regulatory and signaling molecules can be released from the source cells of one type and fuse with the target cells of another type, therefore assisting in the integrated intercellular transfer of information [55]. The content of cargo molecules in exosomes is critically dependent upon their cell origin and physiological or pathological conditions at the moment of exosome formation. Cargo molecules are packed during exosome biogenesis and comprise proteins such as tetraspanins (CD9, CD63, CD81, and CD82), Alix, tumor susceptibility gene 101 (TSG101), Rab GTPases, heat shock proteins (Hsp70, Hsp90), cell adhesion molecules, cytoskeleton proteins, cytokines, regulatory proteins, transcription factors; lipids and genetic cargoes including DNA, mRNA, microRNA, ribosomal RNA (rRNA), circular RNA, long noncoding RNA (lnRNA) (Figure 1) [56,57]. 

Emerging data point to the unique role of exosomes in maintaining tissue and organ homeostasis, as well as recovery after injury. In the nervous system, exosomes are produced by a diversity of cells such as neurons, Schwann cells, microglia, oligodendrocytes, astrocytes, macrophages, and mesenchymal stem cells (MSCs), and they can impact a variety of functions, such as survival and maturation [58].

Mounting evidence points to a putative role of exosomes, secreted by Schwann cells, macrophages, and MSCs in the enhanced regeneration of the peripheral nervous system after injury. In particular, Schwann cell-derived exosomes markedly increased the rate of axon regeneration in vitro and in explant culture of DRG ex vivo, whereas daily administration of the same exosomes into the distal segment of the injured sciatic nerve in vivo resulted in a two-fold increase in axonal growth. Exosomes can change the axon growth-cone morphology to a pro-regenerating phenotype and decrease the activity of the GTPase RhoA inducing the growth-cone collapse and axon retraction. Importantly, this effect was cell-type-specific and was related particularly to Schwann cells, since fibroblast-derived exosomes failed to exert any beneficial actions on axon growth [59]. The exosomes derived from Schwann cells were demonstrated to be packed with growth factors (NGF, BDNF), cytokines (IL-6, IL-8), and cytoskeletal proteins (actin and β3-tubulin) required for the axon growth-cone modulation, myelin-associated glycoprotein, which is in huge demand for axon remyelination, and microRNA being of a paramount importance [60]. 

The abovementioned data point to an active role of exosomes in cell communication and trafficking of a huge variety of biologically active molecules. The molecular content and the ratio of stimulatory and inhibitory molecules inside the exosomes is strictly regulated as well as their targeted delivery to the recipient cells. In addition to biologically active proteins and lipids, DNA and mRNA, exosomes comprise microRNA, which are the well-known regulators of the transcriptional landscape.

### 2.5. The Role of miRNAs in the Regeneration of Peripheral Nerves

MicroRNAs (miRNAs) are short (~22 bp) single-stranded noncoding RNAs which function by a partial binding (partial complementarity) to mRNA. MiRNAs are currently considered to be “master regulators” of gene expression that orchestrate protein expression at the posttranscriptional level by binding to the 3′UTR of target messenger RNA, either hindering mRNA translation or inducing mRNA degradation. For almost 30 years since miRNAs were discovered, they have received widespread attention due to their potential role in a wide variety of physiological and pathological processes, including neurogenesis, neuronal maturation, embryogenesis, and regeneration of the nervous system among other complicated biological processes. MiRNAs can affect the phenotype of both the recipient and donor cells [2,61]. 

Using microarray analysis, Yu et al. demonstrated the dynamic changes in miRNA expression following rat sciatic nerve injury. This group identified 77 regulatory miRNAs obtained from the proximal nerve stumps, which showed significant alteration at four time points after nerve transection. The elevated expression of the miR-221 and miR-222 cluster positively correlated with the injury-induced Schwann cell phenotype. It has been suggested that up-regulated expression of miR-221/222 can stimulate Schwann cell proliferation and migration in vitro by targeting LASS2 (longevity assurance homolog 2), which is a suppressor of cell proliferation; miR-221/222 silencing exerts an opposite effect. In situ hybridization using nerve tissue sections revealed that the increase of miR-221/222 in Schwann cells occurred on the 4th day after the sciatic nerve injury, which is in agreement with the proliferative activity of Schwann cells reaching a peak in 4–11 days after injury in vivo [62].

Tissue plasminogen activator (tPA), a serine protease, is actively involved in nerve regeneration via degrading ECM and cell adhesions and therefore facilitating Schwann cell migration and axon regrowth. It has been found that exosomal miR-340 and tPA were differentially expressed after sciatic nerve injury and negatively correlated with each other. miR-340 regulated tPA through a direct targeting of the 3′UTR of tPA mRNA. The augmented or reduced expression of miR-340 resulted in an increased or decreased tPA secretion, correspondingly, by cultured Schwann cells, their down-regulated fibrinolytic activity and migration. Dysregulation of miR-340 expression at the injury site disrupted cell debris removal and axonal growth [63].

The role of miRNA-1 in nerve regeneration has been clearly demonstrated in vivo, using a rat sciatic nerve injury model in mice. It appears that miRNA-1 directly targets BDNF, resulting in both BDNF mRNA degradation and translation inhibition. Analysis of miRNA-1 expression pattern showed its dynamic time-dependent expression after nerve injury: a rapid decrease within the first days after injury and increase after the 14th day [64]. Similar results on the expression profiles of eight let-7 (lethal-7) miRNAs family members after sciatic nerve injury were obtained by Li et al. [65]. In particular, the temporal change in let-7 miRNA expression in the injured nerve inversely correlated with NGF expression. Let-7 miRNA significantly inhibited proliferation and migration of Schwann cells by targeting NGF mRNA and down-regulating its translation. Inhibition of let-7 miRNAs promoted NGF production by Schwann cells, followed by an elevated axon regrowth in co-culture of primary Schwann cells and DRG explants [65]. These data open a pathway to a therapeutic strategy for peripheral nerve injury, which may overcome the limitations of direct administration of exogenous BDNF or NGF, implicating miRNA-1 or let-7 targeting, correspondingly.

Exploring a possibility of exosome transfer from Schwann cells to regenerating axons, Lopez-Verrilli et al. demonstrated that Schwann cell-secreted exosomes can be specifically internalized by the axons boosting their regeneration in vitro and enhancing regeneration after sciatic nerve injury in vivo. Exosomes modified the growth-cone morphology inducing pro-regenerating phenotype with active filopodia and reduced the activity of the GTPase RhoA, which stimulates the growth-cone collapse and axon retraction [59].

Tissue macrophages can also function as a source of exosomes promoting nerve regeneration. Delivery of M2 macrophage-derived exosomes stimulated the production of NGF and laminin by Schwann cells and enhanced their migration and proliferation in vitro. Exogenous administration of the exosomes to the site of sciatic nerve injury in rats resulted in a 15-fold increase in the number of regenerating axons and up-regulated proliferation of Schwann cells in vivo. The relative expression level of miR-223 was augmented in M2 macrophages and M2-derived exosomes, providing the fundamental molecular mechanisms of nerve regeneration in vivo and in vitro. The transfected M2 macrophages with miR-223 inhibitor exerted an inhibitory effect on Schwann cell migration and proliferation and down-regulated their NGF and laminin production [66]. In addition, exosomes comprising miRNA-21 also contribute to sensory neuron–macrophage interaction after peripheral nerve injury. Sensory neurons produce miR-21-containing exosomes, which can be engulfed by macrophages where miR-21 promotes a pro-inflammatory phenotype [67].

### 2.6. The Role of Exosomes Derived from Mesenchymal Stem Cells (MSCs) in Peripheral Nerve Regeneration

MSCs are self-renewing multipotent progenitors that can be found in various tissues and organs: bone marrow, adipose tissue, dental pulp, umbilical cord blood, etc. An extensive body of literature indicates that MSCs of different origin have the capacity to promote tissue repair and neuroprotection in vivo and in vitro. Specifically, MSCs promote axonal growth, maintain neuronal survival, and significantly improve functional nerve recovery after injury [68,69,70]. 

In vitro experiments have shown that exosomes harvested from MSCs of menstrual fluids or bone marrow promote neurite outgrowth in cortical neuron and sensory neuron cultures. Moreover, the exosomes derived from amniotic MSCs (hAMSCs) exert neuroprotective effects against neuron damage induced by glutamate in vitro, which may be mediated through activating the PI3/K-Akt signaling pathway [71]. The stimulating effect of exosomes derived from gingiva mesenchymal stem cells (GMSC) on peripheral nerve regeneration of crush-injured mice sciatic nerves has been recently demonstrated in vivo in the study by Mao et al. Gelfoam embedded GMSC-derived exosomes delivered locally at the nerve crush injury site facilitated functional recovery and axon regeneration, comparable with the effects observed after the direct transplantation of GMSCs. Mechanistically, GMSC-derived exosomes trigger migration and proliferation of Schwann cells, and increase their expression of protein c-jun, Notch1, GFAP-characteristic genes of pro-regenerative Schwann cell phenotype [72]. 

Besides well-known growth factors such as IGF-1, NGF, HGF (hepatocyte growth factor), VEGF etc. that are incapsulated into exosomes derived from MSCs [53], these exosomes comprise a set of miRNAs that provide regeneration in the nervous tissue. Several publications detailed the miRNA landscape of MSC-derived exosomes and established their biological functions through systemic network analyses using bioinformatic methods and experimental approaches. The sequencing analyses revealed at least 386 miRNAs in MSCs-derived exosomes that target the genes regulating vascular development, capillary-like tube formation, angiogenesis, and cardiovascular system; pathways related to fibrosis (such as Wnt signaling and TGF-β), cell death, cell growth and proliferation, and peripheral nerve regeneration. Therefore, miR-23a-3p, miR-424-5p, miR-144-3p, miR-130-3p, miR-145-5p, miR-29b-3p, miR-29a-3p, miR-25-3p, miR-221-5p, miR-21-5p, miR-125b-5p, miR-22-3p, miR-199a-3p and miR-191-5p affecting more than 60 genes are involved in the regulation of cell growth, miR-199b, miR-218, miR-148a, miR-133b, miR-21 and others, in regulating neuronal differentiation, proliferation, and axonal branching [55,73]. Specifically, it has been shown that MSC-derived exosomes overexpressing miRNA-133b can be captured by the neurons that significantly improve functional recovery after stroke in rats [74]. Once administered into neuronal cell culture medium, these exosomes stimulate neurite growth rate and branching. In particular, miRNA-17-92-containing exosomes of bone marrow MSC origin stimulate the outgrowth of axons of cortical neurons; they exhibit a signaling mechanism relying on PTEN (phosphatase and tensin homolog) suppression, which is a negative regulator of axon growth and neuronal survival. In a similar fashion, miRNA-21 encapsulated into exosomes from the umbilical cord-derived MSC regulates PTEN expression. Exosomal miRNA-146a binds to mRNA encoding epidermal growth-factor receptor (EGFR) in retinal axons and hinders mRNA translation, thus promoting axon regeneration [75].

As described above, inflammation plays an important role in PNS regeneration. It is well-known that MSCs have immunomodulatory properties mostly mediated through paracrine secretion of interferon (IFN)-γ, TGF-β1, HGF, heme oxygenase-1, IL-6, IL-10, prostaglandin E2 (PGE2) and others [76]. Immunological activity of MSC-derived exosomes was discovered by Zhang et al., in 2014 [77]. Since then, exosomes have been implicated in many aspects of immune regulation, such as T-cell proliferation, differentiation of monocytes into dendritic cells, induction of myeloid-suppressive cells, T-regulatory cells, and others. MiRNAs have been involved in the immunomodulatory effects of MSC-derived exosomes as well. These exosomes also contribute to the MSC-mediated immunosuppression of graft-versus-host disease (GVHD) [77]. Additionally, let-7b miRNA cluster present in MSC-derived exosomes contributes to macrophage polarization (from M1 to M2 phenotype) through the TLR4/NF-κB/STAT3/AKT signaling pathway resulting in the inflammatory ablation [78]. Therefore, exosome-unique composition ensuring a wide range of cargoes delivered by these small vesicles to their target cells highlights their remarkable complexity and functional diversity. The latest advances in this field identify the possibility of devising novel therapies based on modified exosomes and their cargoes, which may be a promising strategy for treating peripheral nerve injuries.

## 3. Conclusions

Despite the well-documented regenerative capacity of peripheral nerves to recuperate after injury, the question of full and efficient morphological and functional recovery is far from being resolved. It is especially true after severe injuries and after injuries that are sustained far from the denervated target organs and tissues. 

Recent decades have witnessed remarkable progress towards understanding the mechanistic underpinnings of molecular and cellular processes underlying peripheral nerve regeneration. PNS injury triggers a complex cascade of events inside neuron somas, axons, and the surrounding cells. Schwann cells, fibroblasts, inflammatory cells, trophic and growth factors as well as their receptors, guidance receptors and signaling molecules, exosomes produced by nerve tissue—all these are involved in the nerve regeneration process. The neurons quickly respond to axon injury by altering their transcription program and shifting their phenotype to a regenerative one. In the distal nerve stump, Schwann cells transdifferentiate and function as repair cells. Resident and infiltrating macrophages as well as Schwann cells degrade the debris, facilitating regenerating axons to grow and reinnervate their target organs and tissues. Schwann cells proliferate and migrate to form Büngner bands that guide the growing axons, and subsequently contribute to axon remyelination.

Emerging evidence indicates that miRNA-containing exosomes originating from MSCs, Schwann cells, or macrophages can either stimulate the axonal regeneration directly or do so indirectly by regulating the inflammation response at the site of injury. The identified possibilities of developing novel therapies through making the most of modified exosomes to either activate or regulate the key events in nerve regeneration may open up new avenues in PNS regeneration. Designing new strategies to improve nerve regeneration remains an issue of paramount importance in fundamental research and clinical applications; it may improve patient survival rate and recovery in the near future. 

## Figures and Tables

**Figure 1 ijms-22-13380-f001:**
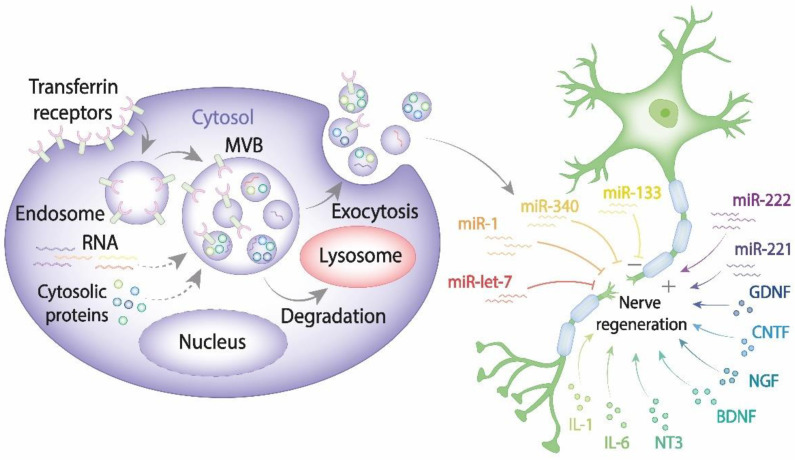
The exosome contribution to nerve regeneration. Exosomes are generated from late endosomes, which are formed by inward budding of the limited multivesicular body (MVB) membrane. The MVBs can either fuse with lysosomes for degradation or with the plasma membrane, therefore releasing exosomes into the extracellular space. Released exosomes taken up by damaged neurons and Schwann cells can either enhance or impair PNS.

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
