# Peer review of "New Frontiers in Peripheral Nerve Regeneration: Concerns and Remedies"

_ijms, 2021, doi:10.3390/ijms222413380_

Round 1

Reviewer 1 Report

1. The paper needs a through revision of its language, not just grammatically, but the language is complicated and needs to be simplified.

2. As it is now the manuscript is far too long and difficult to read. It  needs to be shortened quite a bit to be publishable. In todays world when everyone is busy, we do not need another textbook text about nerve regeneration similar to what we have read before.

3. A very big problem for me is that this review is built almost exclusively on other reviews. The authors have not gone to the original sources for information but just used what others have said about the original papers. You need to go back to the original work and that is what you should refer to in your reference list.

4. I have read many reviews like this and I sadly need to say that I do not think it contributes with any new information or conclusions. If you want to publish this work somewhere, you really need to sharpen it. What is it you want to say with your paper? What in what you write is new and exciting? Why should we read it? What is your new take on peripheral nerve regeneration? Make it snappy, interesting and short.  

5. The pictures do not add to the paper. Make the pictures interesting. As it is now, they are nice, well made drawings but could be found in any school book on nerve regeneration. They do not add new information or conclude new information that you have contributed with. I would like to see pictures that connect to the important new findings or suggestions that you are making in your paper. 

Author Response

The authors would like to thank the Reviewer for his thorough reading and unbiased reviewing of the manuscript, as well as for comments and recommendation, which significantly contributed to the present version of the manuscript.

Please, find our responses to the Reviewer’ comments:

Reviewer comments:

Reviewer #1:

  1. The paper needs a through revision of its language, not just grammatically, but the language is complicated and needs to be simplified.

Response to Reviewer comment: The corrections have been made.

  1. As it is now the manuscript is far too long and difficult to read. It needs to be shortened quite a bit to be publishable. In todays world when everyone is busy, we do not need another textbook text about nerve regeneration similar to what we have read before.

Response to Reviewer comment: The manuscript has been significantly shortened.

  1. A very big problem for me is that this review is built almost exclusively on other reviews. The authors have not gone to the original sources for information but just used what others have said about the original papers. You need to go back to the original work and that is what you should refer to in your reference list.

Response to Reviewer comment: The corrections have been made.

  1. I have read many reviews like this and I sadly need to say that I do not think it contributes with any new information or conclusions. If you want to publish this work somewhere, you really need to sharpen it. What is it you want to say with your paper? What in what you write is new and exciting? Why should we read it? What is your new take on peripheral nerve regeneration? Make it snappy, interesting and short.

Response to Reviewer comment: We made the Review more precise and focused. We considerably reduced it and underscored the most novel data in the field of peripheral nerve regeneration. We have integrated the most recent data on the mechanisms of nerve regeneration mediated my microRNA into the classic concept.

  1. The pictures do not add to the paper. Make the pictures interesting. As it is now, they are nice, well made drawings but could be found in any school book on nerve regeneration. They do not add new information or conclude new information that you have contributed with. I would like to see pictures that connect to the important new findings or suggestions that you are making in your paper.

Response to Reviewer comment: We excluded the first figure that describes a canonical view on the structure of peripheral nerves. The manuscript now comprises the only figure drawn specifically for the review and depicting the most recent findings on the inhibiting and stimulating microRNAs as well as neurotrophins and their role in nerve regeneration.

Reviewer 2 Report

Klimovich and collaborators review molecular and cellular mechanisms for nerve regeneration in the peripheral nervous system by focusing on cell adhesion molecules, extracellular matrix proteins, guidance molecules, neurotrophins, cytokines, exosomes, microRNAs and so on. Although this review article appears to be interesting and is well written, there are many points that should be amended and that may help readers understand this manuscript, as follows:

Major points:

  1. This article has so many abbreviations. The authors should give a list of the abbreviations used.
  2. MSCs are defined as “multipotent stromal cells” in line 561 whereas being as “mesenchymal stem cells” in line 478. Different words should not be given the same abbreviation. If they have the same meaning, a note about this point should be given. Please check once again your manuscript about such points and correct all of them.
  3. The abbreviation for a phrase should be defined when it first appears. In many places, such mistakes can be seen. For example, see “MMP” in lines 165 and 224. Please check your manuscript for such mistakes and correct all of them.
  4. Line 79: not “AMP” but “cyclic AMP”? Does ATF-3 generally mean “activating transcription factor-3”? Please reply to this question.
  5. Line 133: it is not necessary to mention “nerve growth factor” twice. NGF has been already defined in line 102. Please amend these points.
  6. Line 397: please explain “a-ret” shortly.
  7. The letters in Fig. 2 are so small that it is difficult to distinguish them. This figure should be revised to make the letters larger.
  8. Page 11 uses both miRNA and microRNA, which have the same meaning, which is somewhat confusing. Please amend this point.
  9. Line 530: please explain “let-7” shortly.
  10. Line 579: here, PI3/K-Akt is used, while in line 384 PI-3K/Akt is used. Throughout the text, the same abbreviation should be used. Please amend this point.
  11. Line 608: please explain “PTEN” shortly.
  12. Sections 3 and 4 are missing. Please amend this point.
  13. Lines 849 and 850: there are no authors in Reference 85. Please correct this point and check whether all of the references give are right.
  14. The authors mention in this review article a large number of molecules involved in nerve regeneration in the peripheral nervous system. Please check once again if they are stated correctly.

Specific points:

  1. Line 37: not “are” but “is”.
  2. Line 89: do the authors mean “acetylcholine” as “neurotransmitters”? This sentence should be revised.
  3. Line 152: not “FGF” but “bFGF”? Please see line 133. Is “bFGF” different from “FGF2”? Please see line 388. These points should be made clear.
  4. Line 183: please delete one of two “between”.
  5. Line 277: not “contributes” but “contribute”.
  6. Line 297: not “is” but “are”.
  7. Line 326: please expand “FPRL1”.
  8. Line 390: not “CNTF, IL6” but “CNTF and IL6”.
  9. Line 396: not “was” but “were”.
  10. Line 402: please put “.” following [56].
  11. Line 403: not “is” but “are”.
  12. Line 417: “high doses of 20 μg (versus 2 μg)” may be better than “high doses of (20 μg versus 2 μg)”.
  13. Line 422: not “point” but “points”; not “fibres” but “fibers”.
  14. Line 455: not “N-ethylmaleimidesensitive” but “N-ethylmaleimide-sensitive”.
  15. Line 464: not “Hsc70” but “Hsp70”.
  16. Line 486: not “induce” but “induces”.
  17. Line 504: not “has” but “have”.
  18. Lines 504, 541 and 618: not “et al.,” but “et al.”.
  19. Line 520: not “3’-UTR” but “3’UTR”? Please see line 498.
  20. Line 525: not “(miR-1) but “(miRNA-1)”.
  21. Line 526: not “microRNA-1” but “miRNA-1”.
  22. Lines 539 and 623: not “Let-7” but “let-7”?
  23. Line 589: “stromal cell-derived factor-1” may be better than “stromal growth factor-1”.
  24. Please check once again your manuscript in English.

Author Response

The authors would like to thank the Reviewer for his thorough reading and unbiased reviewing of the manuscript, as well as for  comments and recommendation, which significantly contributed to the present version of the manuscript.

Please, find our responses to the Reviewer’ comments:

This article has so many abbreviations. The authors should give a list of the abbreviations used.

Response to Reviewer comment: A list of the abbreviations was added

MSCs are defined as “multipotent stromal cells” in line 561 whereas being as “mesenchymal stem cells” in line 478. Different words should not be given the same abbreviation. If they have the same meaning, a note about this point should be given. Please check once again your manuscript about such points and correct all of them.

Response to Reviewer comment: The corrections have been made. Mesenchymal stem cells are used throughout the manuscript.

The abbreviation for a phrase should be defined when it first appears. In many places, such mistakes can be seen. For example, see “MMP” in lines 165 and 224. Please check your manuscript for such mistakes and correct all of them.

Response to Reviewer comment: The corrections have been made.

Line 79: not “AMP” but “cyclic AMP”? Does ATF-3 generally mean “activating transcription factor-3”? Please reply to this question.

Response to Reviewer comment: Yes, ATF-3 generally mean “activating transcription factor-3”.

Line 133: it is not necessary to mention “nerve growth factor” twice. NGF has been already defined in line 102. Please amend these points.

Response to Reviewer comment: The corrections have been made.

Line 397: please explain “a-ret” shortly.

Response to Reviewer comment: The corrections have been made and this sentence has been changed.

The letters in Fig. 2 are so small that it is difficult to distinguish them. This figure should be revised to make the letters larger.

Response to Reviewer comment: The corrections have been made.

Page 11 uses both miRNA and microRNA, which have the same meaning, which is somewhat confusing. Please amend this point.

Response to Reviewer comment: The corrections have been made.

Line 530: please explain “let-7” shortly.

Response to Reviewer comment: The lethal-7 (let-7) gene was first discovered in the nematode as a key developmental regulator and is linked to apoptosis.

Line 579: here, PI3/K-Akt is used, while in line 384 PI-3K/Akt is used. Throughout the text, the same abbreviation should be used. Please amend this point.

Response to Reviewer comment: The corrections have been made.

Line 608: please explain “PTEN” shortly.

Response to Reviewer comment: The corrections have been made.

Phosphatase and tensin homolog (PTEN) is a phosphatase encoded by the PTEN gene. PTEN acts as a tumor suppression gene and is involved in the regulation of the cell cycle, cell proliferation control. Mutations of this gene associated with many cancers.

Sections 3 and 4 are missing. Please amend this point.

Response to Reviewer comment: The corrections have been made.

Lines 849 and 850: there are no authors in Reference 85. Please correct this point and check whether all of the references give are right.

Response to Reviewer comment: The corrections have been made.

The authors mention in this review article a large number of molecules involved in nerve regeneration in the peripheral nervous system. Please check once again if they are stated correctly.

Specific points:

Line 37: not “are” but “is”.

Response to Reviewer comment: The corrections have been made.

Line 89: do the authors mean “acetylcholine” as “neurotransmitters”? This sentence should be revised.

Response to Reviewer comment: The corrections have been made and the sentence has been revised.

Line 152: not “FGF” but “bFGF”? Please see line 133. Is “bFGF” different from “FGF2”? Please see line 388. These points should be made clear.

Response to Reviewer comment: The corrections have been made.

Line 183: please delete one of two “between”.

Response to Reviewer comment: The corrections have been made.

Line 277: not “contributes” but “contribute”.

Response to Reviewer comment: The corrections have been made.

Line 297: not “is” but “are”.

Response to Reviewer comment: The corrections have been made.

Line 326: please expand “FPRL1”.

Response to Reviewer comment: FPRL1stands for formyl peptide receptor-like 1.

Line 390: not “CNTF, IL6” but “CNTF and IL6”.

Response to Reviewer comment: The corrections have been made.

Line 396: not “was” but “were”.

Response to Reviewer comment: The corrections have been made.

Line 402: please put “.” following [56].

Response to Reviewer comment: The corrections have been made.

Line 403: not “is” but “are”.

Response to Reviewer comment: The corrections have been made.

Line 417: “high doses of 20 μg (versus 2 μg)” may be better than “high doses of (20 μg versus 2 μg)”.

Response to Reviewer comment: We consider the original version “high doses of 20 μg (versus 2 μg)” to be a better option.

Line 422: not “point” but “points”; not “fibres” but “fibers”.

Response to Reviewer comment: The corrections have been made.

Line 455: not “N-ethylmaleimidesensitive” but “N-ethylmaleimide-sensitive”.

Response to Reviewer comment: The corrections have been made.

Line 464: not “Hsc70” but “Hsp70”.  It is Heat shock cognate 71 kDa protein

Response to Reviewer comment: The corrections have been made.

Line 486: not “induce” but “induces”.

Response to Reviewer comment: The corrections have been made.

Line 504: not “has” but “have”.

Response to Reviewer comment: The corrections have been made.

Lines 504, 541 and 618: not “et al.,” but “et al.”.

Response to Reviewer comment: The corrections have been made.

Line 520: not “3’-UTR” but “3’UTR”? Please see line 498.

Response to Reviewer comment: The corrections have been made.

Line 525: not “(miR-1) but “(miRNA-1)”.

Response to Reviewer comment: The corrections have been made.

Line 526: not “microRNA-1” but “miRNA-1”.

Response to Reviewer comment: The corrections have been made.

Lines 539 and 623: not “Let-7” but “let-7”?

Response to Reviewer comment: The corrections have been made.

Line 589: “stromal cell-derived factor-1” may be better than “stromal growth factor-1”.

Please check once again your manuscript in English

Response to Reviewer comment: The corrections have been made.

Round 2

Reviewer 1 Report

I can see that substantial changes have been done, but to sit and read a manuscript with all the changes visible takes time and is difficult. I would like a new manuscript where all the changes have been done. I would also prefer it if the English was better

I still think that the material of the review is good, but as it is now, Iwith all the corrections, I don't want to spend time reading it in depth. Give me a corrected manuscript with proper English and I will read and consider it again.

Author Response

The authors would like to thank the Reviewer for his thorough reading and unbiased reviewing of the manuscript, as well as for comments and recommendation, which significantly contributed to the present version of the manuscript.

Please, find our responses to the Reviewer comments:

Reviewer comments:

I can see that substantial changes have been done, but to sit and read a manuscript with all the changes visible takes time and is difficult. I would like a new manuscript where all the changes have been done. I would also prefer it if the English was better

I still think that the material of the review is good, but as it is now, Iwith all the corrections, I don't want to spend time reading it in depth. Give me a corrected manuscript with proper English and I will read and consider it again.

Response to Reviewer comment: The corrections have been made.

Reviewer 2 Report

This revised manuscript has been largely amended according to my comments, and there is no concern in the present manuscript except for the following just trivial comments:

  1. Line 24: not “Untranslated” but “untranslated”.
  2. Line 29: not “Ciliary” but “ciliary”.
  3. Line 31: not “Deleted” but “deleted”.
  4. Line 37: not “Glial” but “glial”.
  5. Line 38: not “Glia Maturation Factor” but “glia maturation factor”.
  6. Line 50: not “Neural” but “neural”.
  7. Line 51: not “Nerve growth factor, nerve growth factor” but “nerve growth factor”.
  8. Line 58: not “Peripheral” but “peripheral”.
  9. Line 60: not “genes” but “gene”.
  10. Line 64: not “Tissue” but “tissue”
  11. Line 66: not “Uncoordinated” but “uncoordinated”.
  12. Line 132: not “P38” but “p38”.
  13. Line 139: not “RAG” but “RAGs”.
  14. Lines 195 and 196: not “Maturation Factor” but “maturation factor”.
  15. Line 227: not “leads” but “lead”.
  16. Line 249: please add “ECM” to the abbreviation list.
  17. Line 251: not “Ciliary” but “ciliary”.
  18. Line 259: not “activate” but “activates”; not “stimulate” but “stimulates”.
  19. Line 373: please delete one of two periods.
  20. Line 384: not “A-D” but “5A-D”.
  21. Line 594: not “Tumor” but “tumor”.
  22. Line 616: not “peripheral nervous system” but “PNS”.
  23. Line 664: please use either 3’-UTR or 3’UTR (see line 640) throughout the text.
  24. Lines 686 and 764: not “et al.,” but “et al.”.
  25. Line 731: please use either “c-JUN” or “c-Jun” (see line 145).
  26. Line 734: HGF should be defined in this line but not in line 762.
  27. Line 769: not “Let-7b microRNA” but “let-7b miRNA”
  28. It is somewhat difficult to read the whole text because it is a mixture of deleted and added sentences. It will be desirable to check it thoroughly by the authors themselves.

Author Response

The authors would like to thank the Reviewer for his thorough reading and unbiased reviewing of the manuscript.

All the mentioned corrections have been made. 

Round 3

Reviewer 1 Report

Dear Authors,

You have done a very good job revising this manuscript! All credit to you!It is now publishable and all that I said before about it being too long, difficult to read etc. are not applicable any more. Instead it is interesting to read and add to the discussion of nerve regeneration. Very well done.

My only advice to you is getting the English checked again. The manuscript still needs some English editing although this too is much much better. I will accept the manuscript with minor revisions and the minor revisions are only connected to language.